# Biorefinery of Tomato Leaves by Integrated Extraction and Membrane Processes to Obtain Fractions That Enhance Induced Resistance against *Pseudomonas syringae* Infection

**DOI:** 10.3390/membranes12060585

**Published:** 2022-05-31

**Authors:** Fabio Bazzarelli, Rosalinda Mazzei, Emmanouil Papaioannou, Vasileios Giannakopoulos, Michael R. Roberts, Lidietta Giorno

**Affiliations:** 1National Research Council of Italy, Institute on Membrane Technology, CNR-ITM, via P. Bucci 17 C, 87036 Rende, Italy; f.bazzarelli@itm.cnr.it (F.B.); l.giorno@itm.cnr.it (L.G.); 2Engineering Department, Lancaster University, Lancaster LA1 4YW, UK; 3Lancaster Environment Centre, Lancaster University, Lancaster LA1 4YQ, UK; v.giannakopoulos@lancaster.ac.uk (V.G.); m.r.roberts@lancaster.ac.uk (M.R.R.)

**Keywords:** biorefinery, membrane processes, tomato leaves, plant disease, bioactive compounds

## Abstract

Tomato leaves have been shown to contain significant amounts of important metabolites involved in protection against abiotic and biotic stress and/or possessing important therapeutic properties. In this work, a systematic study was carried out to evaluate the potential of a sustainable process for the fractionation of major biomolecules from tomato leaves, by combining aqueous extraction and membrane processes. The extraction parameters (temperature, pH, and liquid/solid ratio (L/S)) were optimized to obtain high amounts of biomolecules (proteins, carbohydrates, biophenols). Subsequently, the aqueous extract was processed by membrane processes, using 30–50 kDa and 1–5 kDa membranes for the first and second stage, respectively. The permeate from the first stage, which was used to remove proteins from the aqueous extract, was further fractionated in the second stage, where the appropriate membrane material was also selected. Of all the membranes tested in the first stage, regenerated cellulose membranes (RC) showed the best performance in terms of higher rejection of proteins (85%) and lower fouling index (less than 15% compared to 80% of the other membranes tested), indicating that they are suitable for fractionation of proteins from biophenols and carbohydrates. In the second stage, the best results were obtained by using polyethersulfone (PES) membranes with an NMWCO of 5 kDa, since the greatest difference between the rejection coefficients of carbohydrates and phenolic compounds was obtained. In vivo bioactivity tests confirmed that fractions obtained with PES 5 kDa membranes were able to induce plant defense against *P. syringae*.

## 1. Introduction

The transition from fossil-based chemicals to the use of bio-based alternatives guided the research towards new cost effective and renewable sources. This means not only a change in raw materials but also the search for new production methods based on a green process approach. 

A substantial amount of biomass (170 billion metric tons (t)) is produced annually in nature through photosynthesis and just 3–4% of it is used for food [1]. Besides, its use in benefits production does not create environmental concerns, since biomass is naturally biodegraded.

Much waste (leaves, wastewater, etc.) is also produced from agricultural activity and food production (about a third), which lead often to serious biomass waste management problems. These wastes are a rich source of carbon-based (carbohydrates, proteins, etc.) substances that if recovered/purified under appropriate cost and environmentally friendly processes can represent an important raw material of high added-value compounds. 

Among the different chemical compounds, 75 % *w*/*w* of the dry biomass is composed of carbohydrates and for this reason, they are currently viewed as the most important feedstock of future green chemistry. These compounds are also gaining much interest due to their involvement in the induction of damage-associated molecular patterns (DAMP) that regulate plant defense signaling, giving them the potential to be used as biopesticides. Various classes of carbohydrates are in fact used as biopesticides as a more natural approach to agriculture, with the aim of replacing highly toxic pesticides, as they can trigger defense responses in plants in a process known as PAMPs (pathogen-associated molecular patterns) [2].

Another important source of bioactive compounds from biomass is the so-called “N-containing fraction” [3] (i.e., protein), which is an important source of amino acids that may have important applications in food and beverages, animal feed, pharmaceuticals, cosmetics, and agrochemicals. In particular, the water-soluble protein fraction (“white fraction”) has attracted much attention for human nutrition, because it is rich in essential amino acids and is very useful in the food sector due to its intrinsic properties (e.g., foaming, emulsifying, and gelling).

Recovery of biophenols from agricultural wastes (e.g., olive mill wastewater, OMWW) and plant biomass (e.g., olive leaves [4,5]) is also of interest, due to their important antioxidant properties. In particular, the development of innovative processes for OMWWs treatment has the dual effect of water purification from biophenols, as they represent a contaminant, and their recovery.

Tomato leaves are generally considered a waste in tomato processing but are a rich source of important compounds, such as biophenols, carbohydrates, etc. [6]. Considering that each tomato plant produces about 0.75 kg of leaf biomass, resulting in about 15 t/ha [7]. The tomato leaf biomass represents an enormous raw material to produce natural important compounds. Besides, cultivated solanaceous crop species including tomato have been shown to contain significant amounts of important metabolites [8,9], which are involved in the protection against abiotic and biotic stresses. However, to produce added-value products from biomass or waste, separation by a sustainable methodology must be considered to develop green production processes. Within the concept of biorefinery, membrane processes have been identified as a key technology for separation [10], extraction, and conversion [11], starting from different biomass sources such as microalgae [12,13], coffee parchment [14], olive leaves for phytotherapic production, etc. [5].Various membrane processes have been developed to concentrate and clarify tomato juice [15,16,17,18] or to extract lycopene from tomato peels [19,20]. However, in order to use biomass that does not compete with food, tomato leaves were used for the first time in this work as a starting material for the production of bioactive molecules. Although membrane processes have widely demonstrated their advantages in the processing of agricultural and food streams, to our knowledge there is little information in the open literature on the type of the membranes, the processes and their performance in the treatment of tomato leaf extracts, and the bioactivity of the processed streams.

For the above-mentioned reasons, this work evaluates the performance of various membranes and their ability to be used in an integrated process to fractionate bioactive compounds from tomato leaves extract, with the final aim to develop an integrated membrane process for producing fractions with biopesticide activity towards the infection of the Gram-negative bacterium *Pseudomonas syringae*. This bacterium can cause necrotic lesions on the leaves, stems, and fruits of tomato, which, without an effective inhibitor of microbial growth, will result in a tremendous economic loss (about 75% yield loss) [21,22].

The challenge is to develop a system capable of recovering bioactive compounds from the original biomass while preserving their structure and function so that their ability to stimulate plant defense is maintained. 

## 2. Materials and Methods

### 2.1. Materials

The Bradford reagent and bovine serum albumin (BSA) were used for protein quantification (Sigma-Aldrich, Milan, Italy). Folin–Ciocalteu’s phenol reagent, sodium carbonate, and gallic acid (Sigma-Aldrich) were used to evaluate biophenols content. Sulfuric acid, phenol, and glucose (Sigma-Aldrich, Milan, Italy) were used to evaluate carbohydrates content. Tomato plants were grown in controlled room environment conditions (12 h light/12 h dark at 22 °C) at Lancaster Environmental Centre. Plants were visually inspected; the defect-free plants were cut and immediately homogenized with 1:2 *w*/*w* H_2_O: plant material with an ordinary blender. This homogeneous mixture was lyophilized and stored in sealed bags at room temperature and in absence of light, to avoid any possible changes in composition. The same conditions were used to grow tomato plants for the subsequent biological testing of the membrane-derived fractions. Fractionation of biomolecules extracted from tomato leaves was performed using commercial membranes of different material (ZrO_2_: zirconia; RC: regenerated cellulose; PAN: polyacrylonitrile; CA: acetate cellulose; hydrophilized PES: polyethersulfone), whose main characteristics are summarized in Table 1.

### 2.2. Methods

#### 2.2.1. Biomolecules and Total Solids Quantification

Folin–Ciocalteau assay was used to evaluate the total content of phenols. Gallic acid was used as a standard within the range of 10–100 mg/L for the calibration curve. Briefly, 0.2 mL of sample was mixed with 1 mL of Folin–Ciocalteau reagent (10 % *v*/*v*). After that, 0.8 mL of sodium carbonate solution (7.5% *w*/*v*) was added. After 30 min of incubation at room temperature, the absorbance of the solution was then measured at 765 nm using a UV–VIS spectrophotometer (Perkin Elmer, lambda EZ, Monza, Italy). 

The phenol–sulfuric acid method was used to determine total carbohydrates. Glucose was used as standard in the range of 10–100 mg/L for the calibration curve. The sample (1 mL) was mixed with 0.5 mL of phenol (5% *w*/*v*) and then 2.5 mL of concentrated sulfuric acid was added. The absorbance of the solution was measured after 20 min at room temperature at 490 nm. 

Proteins were quantified using Bradford protein assay, using bovine serum albumin (BSA) as standard in the range of 2–20 mg/L for the calibration curve. The test was performed by mixing 1 mL of the sample with 1 mL of Bradford reagent. The absorbance of the solution was measured at 595 nm after 30 min at room temperature. 

#### 2.2.2. Production of Aqueous Extract from Lyophilized Tomato Leaf 

The extraction of biomolecules from the homogenized freeze-dried tomato leaves was tested by using different liquid/solid ratios (L/S) (15–40), temperatures (25–60 °C), and pH (2.3–7) with water as the extraction solvent. The suspension was stirred for 2 h and then sonicated (power of 120 W and a frequency of 45 kHz) for 20 min. After this process, the supernatant was recovered by centrifugation (9000 rpm for 10 min) and treated by membrane processes. 

#### 2.2.3. Biopesticide Activity of the Collected Fractions

Tomato plants were grown for three weeks under control environment conditions (12 h dark/12 h light at 22 °C). The fractionated extracts were diluted to a final carbohydrate concentration of 1 mg/mL and applied to a tomato leaf by spraying, two days prior to the infection with *P. syringae* for inducing the plant immune response. This leaf was appropriately marked to be used in the next infection step. This plant leaf after 2 days was infected with *P. syringae* by immersing it (20 s) in a suspension of 1 × 10^8^ CFU/mL in 10 mM MgCl_2_, 0.2% *v*/*v* rifampicin, and 0.05% *v*/*v* Silwet L-77. Then the plants were left growing for two more days. The infected leaf was removed from each plant, weighed, and homogenized in a mortar with 10 mM MgCl_2_ at a ratio of 1:10 *w*/*v*. Ten microliters of the homogenized leaf material supernatant was taken and diluted 10-fold before it was plated on agar with 0.2% *v*/*v* rifampicin. The plates were incubated for 30 hours at 28 °C and the colonies were counted. 

#### 2.2.4. Membrane Process Equipment 

The filtration experiments were performed in cross-flow mode, by using two different stages. In the first stage, UF membranes of 30 and 50 kDa of different materials were used, while in the second stage membranes with NMWCO of 5, 2, and 1 kDa were tested (Table 1). In both cases, the membrane was allocated in a stainless-steel cross-flow module. The system is also composed of a feed and permeate tank, a peristaltic pump (Masterflex L/S, General Control S.p.A., Milan, Italy) for the first membrane stage, and a magnetic drive gear pump (Micropump GC series, Techma GPM S.r.l., Milan, Italy) for the second stage. Two manometers (0–6 bar) for measuring the inlet and outlet pressures were placed before and after the membrane cell to measure transmembrane pressure (TMP) (see Appendix A). The axial flow rate used in the first stage was 15 L/h (TMP 0.5 bar), while the axial flow rate used in the second stage was 50 L/h (TMP: 3 bar for filtration with 5 kDa membranes; TMP: 5 bar for filtration with 1 and 2 kDa membranes).

After filtration of aqueous leaves extract, the fouled membranes were rinsed with deionized water at room temperature and subsequently cleaned with NaOH (0.1% *w*/*v* aqueous solution) at 40 °C for 30 min, applying a TMP of 0.5 and 3 and 5 bar for 30–50 kDa and 1–5 kDa membranes, respectively. The pure water permeance of membranes was measured before and after the two cleaning steps to evaluate the fouling index and cleaning efficiency.

The fouling index (FI) was determined according to Equation (1).
(1)$$\mathrm{FI}=\left(1-\frac{\mathrm{WP}1}{\mathrm{WP}0}\right)\times 100$$
where WP0 and WP1 are the pure water permeance (L/h·m^2^·bar) of the membrane before and after the treatment of the tomato leaves extract, respectively.

The cleaning efficiency (CE) was evaluated according to the following Equation (2):
(2)$$\mathrm{CE}=\left(\frac{\mathrm{WP}2}{\mathrm{WP}0}\right)\times 100$$
where WP2 is the water permeance of the membrane after chemical cleaning.

The experiments were carried out in concentration mode, so the volume reduction factor (VRF) was calculated from Equation (3).
(3)$$\mathrm{VRF}=\left(\frac{\mathrm{Vf}}{\mathrm{Vr}}\right)$$
where Vf and Vr are the feed and retentate volume, respectively.

The membrane rejection (Rj) towards biomolecules was evaluated according to the following Equation (4): (4)$$\mathrm{Rj}=\left(1-\frac{\mathrm{Cp}}{\mathrm{Cf}}\right)$$
where Cp and Cf are the concentration of biomolecules in the permeate and feed stream, respectively.

Membrane wettability was measured by static water contact angle measurement (CAM 200 instrument, KSV Instruments, Ltd., Helsinki, Finland). Briefly, water (5 μL) was applied to the membrane with an automatic microsyringe, and each measurement was repeated five times on different membrane pieces.

## 3. Results

### 3.1. Production of Tomato Leaves Extract

To determine the best conditions for biomolecules extraction, experiments were carried out at various liquid/solid (L/S) ratios, temperatures, and pH values (Figure 1). The amount of proteins, biophenols, and carbohydrates per g of initial leaves weight extracted in water by varying the water/solid ratio is reported in Figure 1a. Extraction time (2 h) and temperature (37 °C) were kept constant. This parameter permits minimizing water consumption, in line with the development of green processes. For all values of the L/S ratio, a higher amount of carbohydrates with respect to the other biomolecules was observed.

The amount of carbohydrates extracted decreased with the L/S ratio increase, reaching the maximum when an L/S of 20 was used. This is the reason why this value was used for further investigation. The amount of biophenols and proteins is almost constant at all the L/S values tested, but it is particularly lower with respect to carbohydrates at an L/S of 20 (less than 95% for protein and 90% for biophenols, respectively), which is very important in case of fractionation since the compound of interest is present in a mixture with a lower amount of impurities. 

The effect of temperature (25–60 °C) on the extraction of biomolecules was also studied (Figure 1b), as this parameter can increase the extraction of molecules, but it can also cause denaturation phenomena of labile compounds. The concentration of carbohydrates increases by about 45% from 25 to 37 °C, where the highest extraction was obtained (0.064 g_carbohydrates_/g_leaves_); a decreasing trend was observed after 37 °C. This is probably due to the fact that high temperatures can damage labile biomolecules. When carbohydrates were extracted from other biomasses (e.g., leaves of Morus alba L. [27] and Astragalus cicer L. [28]), a higher temperature (70–80 °C) was required for stronger extraction due to the greater ability of the solvent to dissolve the compounds [29]. However, the higher temperature may be necessary due to the different starting material and the different particle size of the leaves before extraction. Nevertheless, there are also examples of high carbohydrate extraction at a temperature between 40–60 °C, confirming that the best temperature for carbohydrate extraction is closely related to the plant starting material. 

The pH of the extraction solution is another important parameter to consider as it can affect the solubility and stability of the biomolecules. Figure 1c shows the amount of extracted biomolecules in relation to the amount of freeze-dried leaves as a function of pH (2–7). The highest amount of carbohydrates (0.1 g/g_leaves_) was obtained at pH 2.3. This result is consistent with other studies in which an acidic pH favors hydrolysis of the insoluble form of carbohydrates by increasing their extraction. In particular, the higher hydrogen concentration at the mentioned pH promotes the hydrolysis of the insoluble polysaccharides and their release from the cell wall, as previously reported [30]. Besides, a pH of around 2 avoids phenols oxidation and mildew growth [31]. Based on the obtained results, the extract subsequently treated by membrane processes was the one obtained at 37 °C, pH of 2.3, and using an L/S ratio of 20.

### 3.2. Ultrafiltration (UF) of Aqueous Extract

After extraction of the biomolecules, the solution was treated with membrane systems, using 30–50 kDa and 1–5 kDa membranes for the first and second stages, respectively. The aim of the first stage was to remove proteins from the aqueous extract, while the second stage aimed to preliminarily fractionate carbohydrates and biophenols. For this purpose, a first study was performed to select membrane material with a low fouling index.

#### 3.2.1. Selection of Membrane Material for Protein Removal from Tomato Leaves Extract

The removal of proteins from the aqueous extract using membranes with different physicochemical properties (Table 1) was carried out. The selection of the best membrane considering permeate flux, fouling index, and membrane rejection towards bioactive compounds (carbohydrates, proteins, biophenols) was performed. Figure 2 shows the constant permeate flux (VRF: 2) of aqueous extract ultrafiltration using regenerated cellulose (RC, 30 kDa), polyacrylonitrile (PAN, 30 kDa), and zirconia UF (ZrO_2_, 50 kDa) membranes. The use of both polymeric membranes resulted in a 66% higher permeate flux compared to the ceramic membrane (Figure 2). Although the cleaning efficiency is of the same order of magnitude (Figure 3), a lower fouling index (14 vs. 82%) was obtained in the case of RC compared to the other two membranes. This indicates that the decrease in water permeance (from 385 L/h·m^2^·bar to about 330 L/h·m^2^·bar) observed for the RC membrane was mainly caused by reversible fouling phenomena. The lower fouling tendency is due to the high hydrophilicity of the RC membrane (water contact angle: 19°±4 [24]) with respect to the other membranes (Table 1), which favors low fouling due to protein adsorption, as previously shown by [13,32] using other types of biomass as starting material.

In Figure 4, the membrane rejection towards the main biomolecules, present in the feed, was reported. All membranes tested showed higher rejection of proteins (74% for PAN membranes, 77% for ceramic membranes, and more than 85% for RC membranes) and lower rejection of biophenols and carbohydrates (less than 15%). However, on the basis of the higher flux and low tendency to fouling, RC membranes were selected for the separation of proteins from biophenols and carbohydrates.

Figure 5 shows the time evolution of the flux and VRF through the RC membrane using the aqueous extract of tomato leaves. The initial permeate flux of about 45 L/h·m^2^ gradually decreased within 50 min, reaching the constant flux (10 L/h·m^2^) after about 200 min of operation in concentration mode. The decrease in permeate flux with VRF increasing can be attributed to both fouling phenomena and increased concentration of proteins in the retentate.

After a VRF of about 3, 85% of proteins were retained by the membrane, while a low rejection of the membrane to carbohydrates and biophenols (less than 10%) was obtained. The initial water permeance was almost restored, even after three different UF processes in concentration mode, further confirming the low fouling tendency of the used membrane.

#### 3.2.2. Selection of Membrane Material for Carbohydrates Fractionation

In order to select the most suitable membrane for further fractionation of the biomolecules, the permeate collected in the first step with the RC membrane was then subjected to a second membrane step with a smaller pore size. In particular, membranes with NMWCO of 1, 2, and 5 kDa were used.

Figure 6 shows the constant permeate flux of all the tested membranes obtained with a VRF of 2. Higher permeate flux was achieved by using 5 kDa ceramic membranes and 2 kDa CA membranes, followed by 5 and 1 kDa PES, respectively. Membranes with the same NMWCO (5 kDa PES and ZrO_2_) (Figure 7) showed a comparable tendency to foul, consistent with their hydrophilicity (water contact angle in Table 1). In contrast, analysis of membranes made of the same material (same roughness, 5 kDa: 1.59 nm [25] and 1 kDa: 1.30 nm [26]) but with different NMWCO (PES membranes) showed a higher FI for the 1 kDa membrane, in agreement with its lower hydrophilicity (Table 1), resulting in a higher fouling tendency.

Consistent with the flux behavior, the CA membranes showed lower FI (Figure 7) because they have a lower chemical interaction with biomolecules due to a different membrane material and surface chemistry, resulting in a lower fouling tendency. However, using the cleaning method described in M-M, a cleaning efficiency of more than 90% was obtained for all membranes tested. Figure 8 shows the membranes’ rejection to carbohydrates and biophenols. Comparable rejections of tomato carbohydrates (mainly soluble carbohydrates such as sucrose, glucose, fructose, and oligogalacturonides [33,34]) were observed using membranes with the same NMWCO (5 kDa ceramic and PES membranes), but higher rejection of biophenols (more than 50%) was obtained with PES membranes. This result confirms that the observed rejection is due to phenomena other than size exclusion alone, such as the chemical interaction between solute/membrane and solute/solute, as previously observed by Galanakis et al. [35] and Conidi et al. [36]. Since biophenols in tomato leaves (mainly gallic acid, chlorogenic acid, caffeic acid, ferulic acid, rutin, and quercetin [37]) are amphiphilic compounds consisting of a hydrophobic aromatic group and a hydrophilic hydroxyl group, they can interact with the PES membrane through both hydrophobic and polar interactions, resulting in higher rejection compared to carbohydrates [38]. 

This was also confirmed when the rejection between membranes with the same material (PES) but different NMWCO was considered. Under the above conditions, by reducing the NMWCO from 5 to 1 kDa, an increase in membrane rejection of about 10 percentage points towards both biomolecules was observed. In this particular case, the material is the same, so the nature of the physical–chemical interactions is the same and, as expected, the observed rejection is higher for the lower NMWCO membrane. Similar rejection results against carbohydrates and biophenols were obtained by Conidi et al. [39], using 1 kDa polyamide and polyethersulfone membranes, but starting from another solanaceous crop extract (Goji leaves). When filtrated with the CA membrane, the same rejection was observed for both categories of biomolecules, showing lower membrane selectivity for the treated compounds and further confirming the role of membrane material in the separation.

Based on the membrane rejection results, the fractions obtained after 5 kDa PES UF were considered for further characterization, because they showed the greatest difference between the rejection coefficients of carbohydrates and phenolic compounds. These fractions were used to treat the leaves of tomato plants two days prior to inoculation with the bacterial pathogen, *P. syringae*. Table 2 shows *P. syringae* colony-forming units on leaves after three from inoculation. Compared to the control, all processed samples showed a decrease in *P. syringae* population growth of about one order of magnitude, demonstrating the presence of compounds able to promote disease resistance in tomato plants. A comparable decrease in *P. syringae* leaf population was observed when the plant was treated with retentate and permeate, as both fractions were sprayed to the leaves at a fixed and equal carbohydrates concentration (1 mg/mL). This result suggests that molecules with a molecular weight (MW) of less than 5 kDa (permeate) play a role in the induction of plant resistance to *P. syringae* infection.

## 4. Conclusions

In this work, the potential of a new process for the fractionation of high-value compounds from tomato leaves by combining aqueous extraction and membrane processes was studied. Tomato leaf extract obtained at 37 °C with a pH of 2.3 and an L/S of 20 resulted in higher extraction of the most abundant biomolecules, namely the carbohydrates. 

The aqueous extract was then processed by membrane technology, using 30–50 kDa and 1–5 kDa membranes for the first and a second stage, respectively. In particular, the permeate from the first stage, which was used to remove proteins, was treated in the second stage to select the most suitable membrane material for further fractionation of carbohydrates and biophenols.

For the first stage, regenerated cellulose gave the best results in terms of protein removal (85%) and fouling index (14%). Among the membranes tested in the second stage, PES of 5 kDa showed the highest rejection of biophenols (53%) and the lowest for carbohydrates (20%).

Preliminary results in *P. syringae*-infected plants treated with fractions obtained with 5 kDa PES membranes showed a significant decrease (one order of magnitude) in bacterial population growth compared to the control. In addition, the similar activity between retentate and permeate indicates that compounds with an MW lower than 5 kDa may play a role in inducing plant resistance.

## Figures and Tables

**Figure 1 membranes-12-00585-f001:**
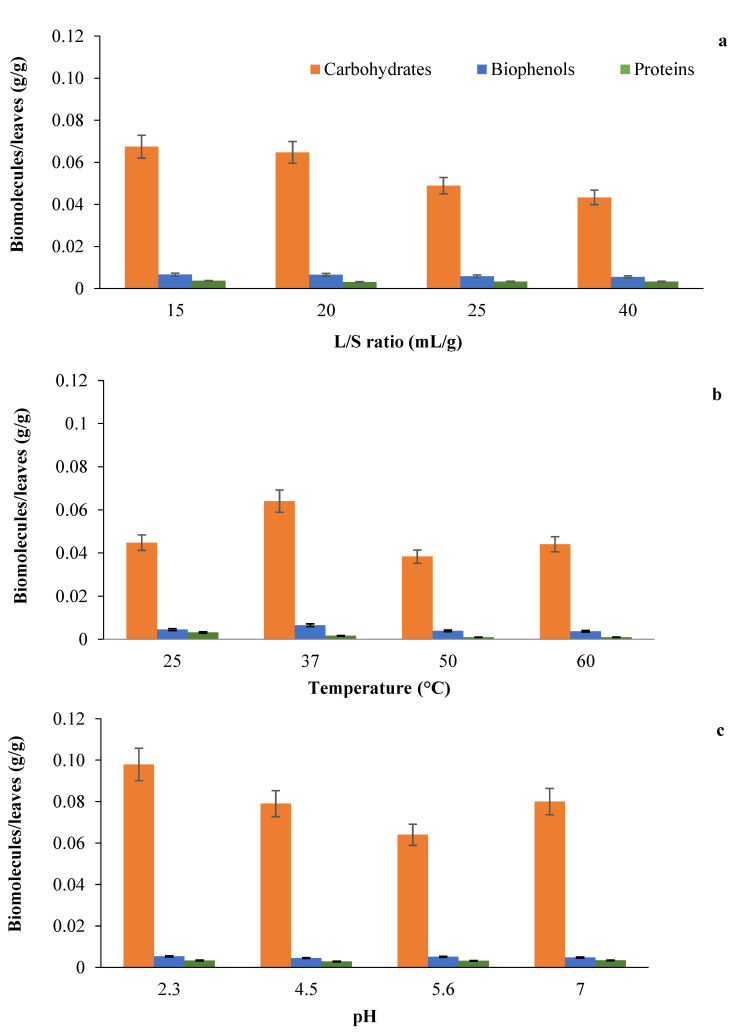
Extraction of biomolecules (carbohydrates, biophenols and proteins) as a function of L/S ratio (**a**), (T = 37 °C, pH = 5.6); temperature (**b**), (L/S ratio = 20, pH = 5.6); and pH (**c**) (T = 37 °C, L/S ratio = 20).

**Figure 2 membranes-12-00585-f002:**
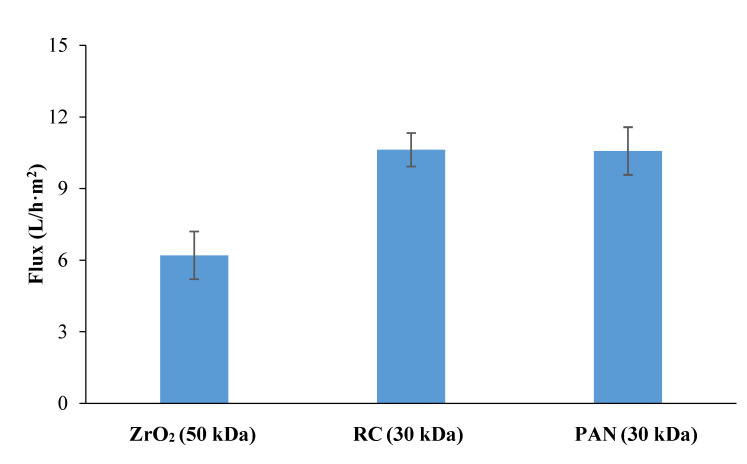
Constant permeate flux (VRF: 2) of tomato aqueous extract through 50 kDa zirconia (ZrO_2_), 30 kDa RC, and 30 kDa PAN membranes.

**Figure 3 membranes-12-00585-f003:**
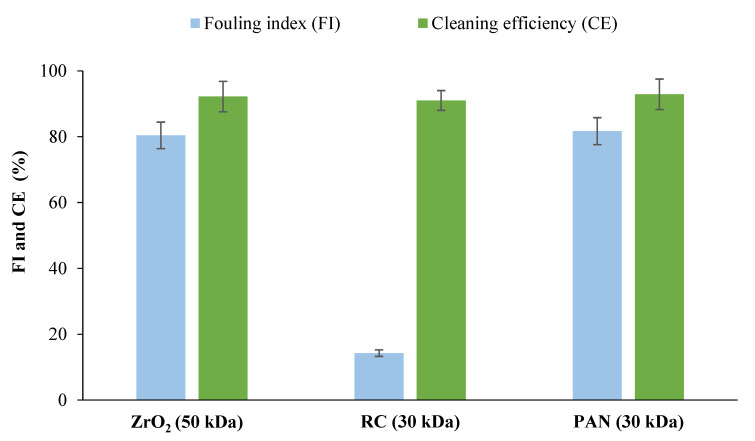
Fouling index and cleaning efficiency of tested membrane.

**Figure 4 membranes-12-00585-f004:**
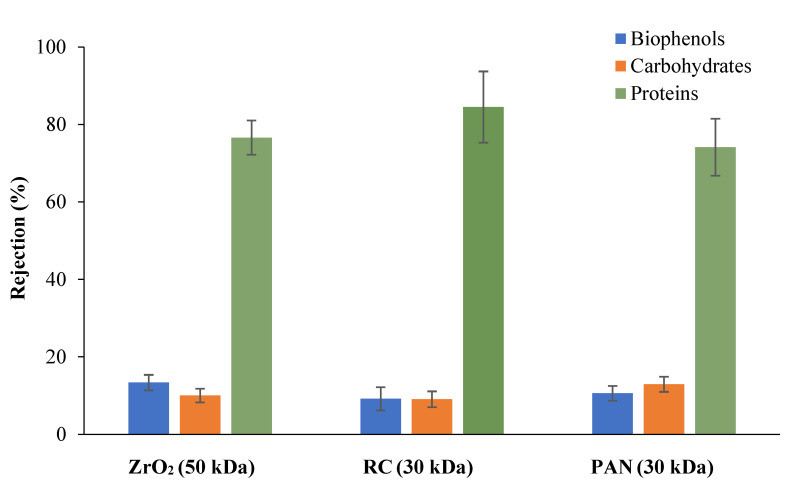
Rejection of membranes towards main biomolecules.

**Figure 5 membranes-12-00585-f005:**
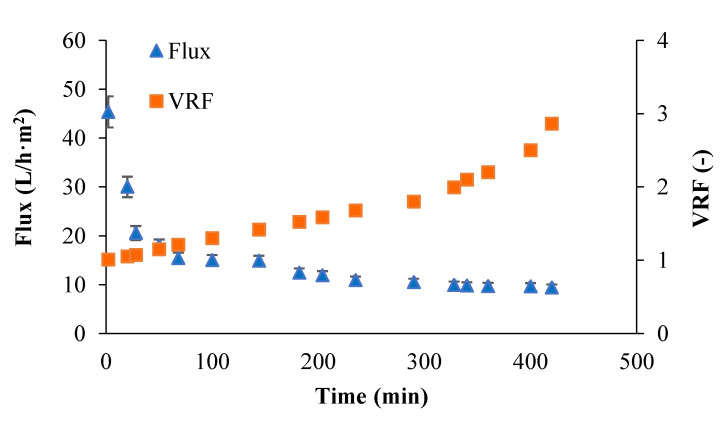
Permeate flux and VRF obtained during the filtration of tomato leaves aqueous extract by RC membrane as a function of time.

**Figure 6 membranes-12-00585-f006:**
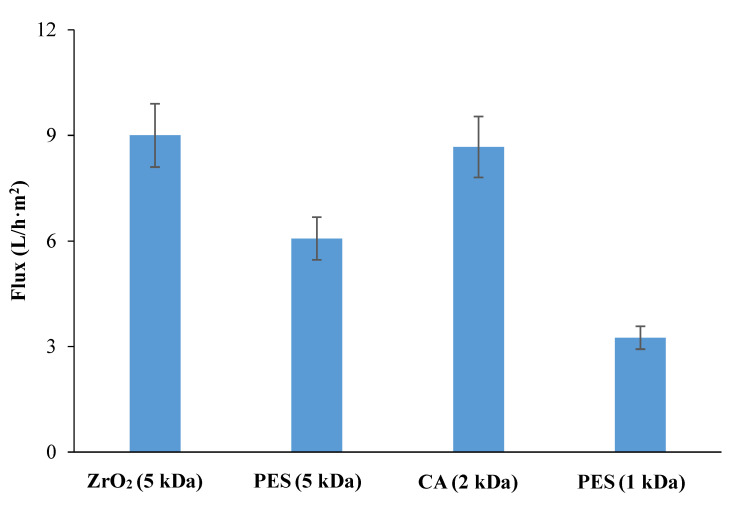
Constant permeate flux (VRF: 2) obtained by using ceramic (ZrO_2_), polyethersulfone (PES) and cellulose acetate (CA) membranes.

**Figure 7 membranes-12-00585-f007:**
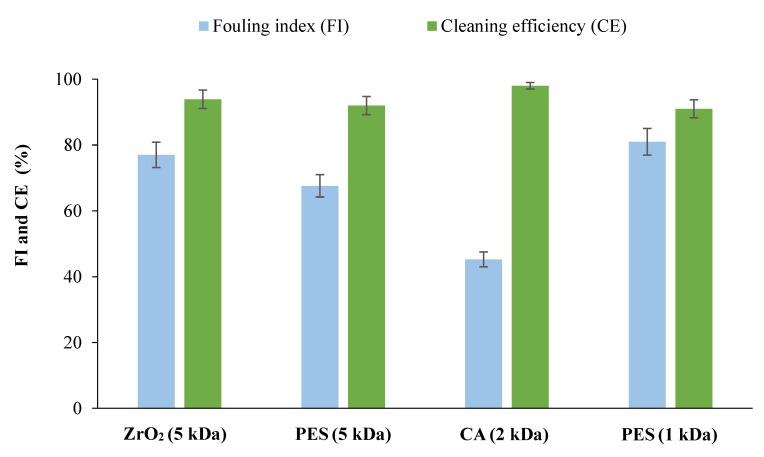
Fouling index and cleaning efficiency after the filtration with all the tested membranes.

**Figure 8 membranes-12-00585-f008:**
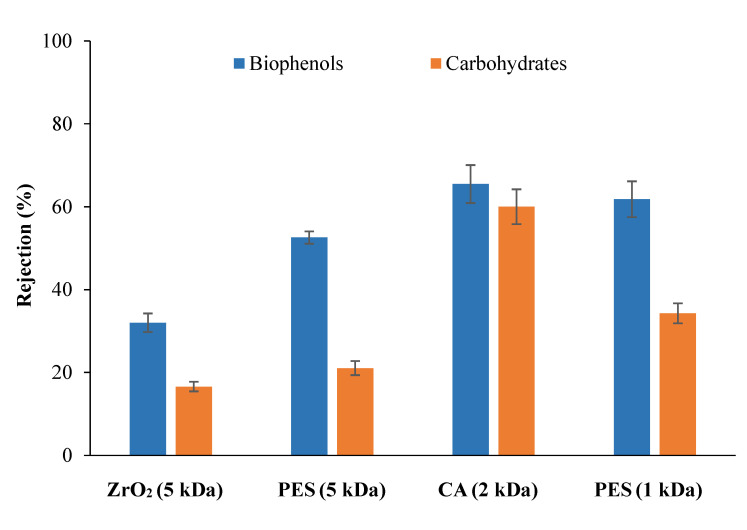
Rejection of tested membranes towards biophenols and carbohydrates.

**Table 1 membranes-12-00585-t001:** Characteristics of selected membranes.

Membrane Type	ZrO_2_	RC	PAN	PES	ZrO_2_	CA	PES
Manufacturer	Tami	Merck Millipore	GE Osmonics	Microdyn- Nadir	Tami	Microdyn-Nadir	Microdyn-Nadir
Configuration	tubular	flat	flat	flat	tubular	flat	flat
NMWCO (kDa)	50	30	30	5	5	2	1
Membrane surface area (cm^2^)	42	12.6	12.6	12.6	42	12.6	12.6
Water contact angle (°)	42.0 ^a^	19.0 ^b^	34.5 ^c^	54.3 ^d^	58.0 ^a^	71.8 ^c^	72.0 ^e^

ZrO_2_: zirconia; RC: regenerated cellulose; PAN: polyacrylonitrile; CA: cellulose acetate; hydrophilized PES: polyethersulfone. NMWCO: nominal molecular weight cut-off. a: [23]; b: [24]; c: our measurements; d: [25] e: [26].

**Table 2 membranes-12-00585-t002:** Biological activity of the fractions collected after filtration by 5 kDa PES membranes.

Samples	*P. syringae* Leaf Population (CFU/mL)
Control	>7 × 10^5^
Feed	3.9 ± 2.1 × 10^4^
Permeate	1.9 ± 0.1 × 10^4^
Retentate	1.5 ± 0.3 × 10^4^

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
