# Peer review of "Biorefinery of Tomato Leaves by Integrated Extraction and Membrane Processes to Obtain Fractions That Enhance Induced Resistance against Pseudomonas syringae Infection"

_membranes, 2022, doi:10.3390/membranes12060585_

Round 1
Reviewer 1 Report
Review of membranes-1735547
This is an interesting manuscript about processing of tomato leaves via extraction and downstream processing using membranes, as well as the exploration about their microbiological activity. This manuscript will be a nice contribution to the knowledge of membrane science, (bio)chemical engineering, vegetal physiology, etc. There are some aspects that have to be revised first, as follows:
- Please kindly follow the MDPI template, especially for inserting tabs to indicate new paragraphs.
- For the collected extracts, perhaps it would be even impactful if the extracts are characterized and their molecular structures elucidated (or predicted), especially for the biophenols or polyphenols part.
- Please kindly write units in X/Y format only, or X Y-1 format only, but not both. Most of the units in this manuscript are in X/Y format, but in line 67, there is a “t ha-1”. The X Y-1 format is more suitable when we have multiple variables in the denominator, e.g. flux, L m-2 h-1 or sometimes L m-2 h-1 bar-1. But for this case, just simply revise the unit in line 67.
- Please merge these short paragraphs (line 70-72, and line 73-74) with line 65-69.
- Please merge this short paragraph (line 75-78) with line 79-83.
- Line 83: Please provide more description about Pseudomonas syringae. Is it Gram positive or Gram negative, what kind of damage it brings, how severe its impact (e.g. harvest failure of XYZ million ton globally), etc.
- Line 93: …homogenized… --> Please use British English only, or American English only, but not both.
- Line 98: ZrO2 --> with subscripted 2
- Table 1: Please check the MDPI template for font type, font size, spacing, etc. for constructing a table.
- Line 126: Please separate “2” and “h” with a space.
- Line 132: 22 °C --> please follow the correct writing of the unit of temperature in line 141, not with lowercase o, uppercase O, or number 0.
- Line 136: 1×108 --> with superscripted 8, and multiplication sign ×
- Line 137: MgCl2 --> with subscripted 2
- Line 139: MgCl2 --> with subscripted 2
- Line 161, 166, 171, 175: Equation, with uppercase E.
- Line 185: Please separate “2” and “h” with a space.
- Please merge this short paragraph (line 188-189) with the previous paragraph or the next one.
- Line 203: Morus alba --> scientific name must be written in italic.
- Please merge this short paragraph (line 218-219) with the previous paragraph.
- Figure 1a, 1b, 1c and their caption: Please rearrange them to fit a single page, thank you.
- Line 251, caption Figure 1b: Please delete the extra dot between “pH =” and number 5
- Please merge this short paragraph (line 276-277) with the previous paragraph or the next one.
- Line 291: ZrO2 --> with subscripted 2
- Line 305: m2 --> with superscripted 2
- Figure 5 and its caption: Please rearrange them to fit a single page, thank you.
- Please merge this short paragraph (line 323-324) with the next paragraph.
- Please merge these short paragraphs (line 330, and line 331-333) with the next paragraph.
- Line 353: ZrO2 --> with subscripted 2
- Please merge this short paragraph (line 377-378) with the previous paragraph.
- Reference 7: ACS Omega --> uppercase O
- References 15, 16, 17, 22: Scientific names must be written in italic
- Reference 23: Please change “WettiGG” with “WettiNG”. The web version of this article indeed has an error, but the scanned PDF of the printed version is correct.
Author Response
REVIEWER 1
This is an interesting manuscript about processing of tomato leaves via extraction and downstream processing using membranes, as well as the exploration about their microbiological activity. This manuscript will be a nice contribution to the knowledge of membrane science, (bio)chemical engineering, vegetal physiology, etc. There are some aspects that have to be revised first, as follows:
- Please kindly follow the MDPI template, especially for inserting tabs to indicate new paragraphs.
ANSWER: DONE
- For the collected extracts, perhaps it would be even impactful if the extracts are characterized and their molecular structures elucidated (or predicted), especially for the biophenols or polyphenols part.
ANSWER: Thank you, this is a good suggestion! We have added the main phenolic compounds in tomato leaves and we have discussed on how they can interact with the membrane in case of higher rejection on the basis of their main structure. Besides we have also included new references.
In the following the modified sentence (page 11 line 386) and the new reference:
“Since biophenols in tomato leaves (mainly gallic acid, chorogenic acid, caffeic acid, ferulic acid, rutin and quercetin [37]) are amphiphilic compounds consisting of a hydrophobic aromatic group and a hydrophilic hydroxyl group, they can interact with the PES membrane through both hydrophobic and polar interactions, resulting in higher rejection compared to carbohydrates. [38]. “
The following references was also added
Silva-Beltrán, N. P., Ruiz-Cruz, S., Chaidez, C., Ornelas-Paz, J. D. J., López-Mata, M. A., Márquez-Ríos, E., & Estrada, M. I. (2015). Chemical constitution and effect of extracts of tomato plants byproducts on the enteric viral surrogates. International Journal of Environmental Health Research, 25(3), 299-311.
For what concern carbohydrates in tomato leaves we have added the following sentence (page11 line 379 ) and two new references in the following reported : “
“Comparable rejections of tomato carbohydrates (mainly soluble carbohydrates such as sucrose, glucose fructose and oligogalacturonides)….”
- Please kindly write units in X/Y format only, or X Y-1 format only, but not both. Most of the units in this manuscript are in X/Y format, but in line 67, there is a “t ha-1”. The X Y-1 format is more suitable when we have multiple variables in the denominator, e.g. flux, L m-2 h-1 or sometimes L m-2 h-1 bar-1. But for this case, just simply revise the unit in line 67.
ANSWER: DONE
- Please merge these short paragraphs (line 70-72, and line 73-74) with line 65-69.
ANSWER: DONE
- Please merge this short paragraph (line 75-78) with line 79-83.
ANSWER: DONE
- Line 83: Please provide more description about Pseudomonas syringae. Is it Gram positive or Gram negative, what kind of damage it brings, how severe its impact (e.g. harvest failure of XYZ million ton globally), etc.
ANSWER At the end of the introduction the following sentence was added with the requested info:
“the gram-negative bacterium Pseudomonas syringae. This bacterium can cause necrotic lesions on the leaves, stems, and fruits of tomato, which, without an effective inhibitor of microbial growth, will result in a tremendous economic loss (about 75% yeld loss)”.
Besides, the following references were added: Singh, V. K., Singh, A. K., & Kumar, A. (2017). Disease management of tomato through PGPB: current trends and future perspective. 3 Biotech, 7(4), 1-10.
Lanna Filho, R., De Souza, R. M., Ferreira, A., Quecine, M. C., Alves, E., & De Azevedo, J. L. (2013). Biocontrol activity of Bacillus against a GFP-marked Pseudomonas syringae pv. tomato on tomato phylloplane. Australasian Plant Pathology, 42(6), 643-651.
- Line 93: …homogenized… --> Please use British English only, or American English only, but not both.
ANSWER: DONE
- Line 98: ZrO2 --> with subscripted 2
ANSWER: DONE
- Table 1: Please check the MDPI template for font type, font size, spacing, etc. for constructing a table.
ANSWER: DONE
- Line 126: Please separate “2” and “h” with a space.
ANSWER: DONE
- Line 132: 22 °C --> please follow the correct writing of the unit of temperature in line 141, not with lowercase o, uppercase O, or number 0.
ANSWER: DONE
- Line 136: 1×108 --> with superscripted 8, and multiplication sign ×
ANSWER: DONE
- Line 137: MgCl2 --> with subscripted 2
ANSWER: DONE
- Line 139: MgCl2 --> with subscripted 2
ANSWER: DONE
- Line 161, 166, 171, 175: Equation, with uppercase E.
ANSWER: DONE
- Line 185: Please separate “2” and “h” with a space.
ANSWER: DONE
- Please merge this short paragraph (line 188-189) with the previous paragraph or the next one.
ANSWER: DONE
- Line 203: Morus alba --> scientific name must be written in italic.
ANSWER: DONE
- Please merge this short paragraph (line 218-219) with the previous paragraph.
ANSWER: DONE
- Figure 1a, 1b, 1c and their caption: Please rearrange them to fit a single page, thank you.
ANSWER: DONE
- Line 251, caption Figure 1b: Please delete the extra dot between “pH =” and number 5
ANSWER: DONE
- Please merge this short paragraph (line 276-277) with the previous paragraph or the next one.
ANSWER: DONE
- Line 291: ZrO2 --> with subscripted 2
ANSWER: DONE
- Line 305: m2 --> with superscripted 2
ANSWER: DONE
- Figure 5 and its caption: Please rearrange them to fit a single page, thank you.
ANSWER: DONE
- Please merge this short paragraph (line 323-324) with the next paragraph.
ANSWER: DONE
- Please merge these short paragraphs (line 330, and line 331-333) with the next paragraph.
ANSWER: DONE
- Line 353: ZrO2 --> with subscripted 2
ANSWER: DONE
- Please merge this short paragraph (line 377-378) with the previous paragraph.
ANSWER: DONE
- Reference 7: ACS Omega --> uppercase O
ANSWER: DONE
- References 15, 16, 17, 22: Scientific names must be written in italic
ANSWER: DONE
- Reference 23: Please change “WettiGG” with “WettiNG”. The web version of this article indeed has an error, but the scanned PDF of the printed version is correct.
ANSWER: DONE

Reviewer 2 Report
The manuscript deals with the study of the performance of various commercially produced ultrafiltration membranes and their ability to be used in an integrated process to fractionate bioactive compounds from tomato leaves extract. The paper is logically structurized and clearly presented. However, it resembles more technical report rather than a scientific publication. Fouling phenomena have to be discussed in more detail taking into account membrane hydrophilic-hydrophobic properties and surface roughness parameters.
Comments
1. State, please, clearly what is the novelty of the study. Cite and analyze, please, more references on membrane-based processes for extraction of valuable compounds from tomatos or cultivated solanaceous crop species.
2. Why pH 2.3 is preferable for the extraction?
3.Determine, please, and compare membrane water contact angles and use these data to discuss membrane fouling. Is PES membranme hydrophilized or not?
Author Response
REVIEWER 2
The manuscript deals with the study of the performance of various commercially produced ultrafiltration membranes and their ability to be used in an integrated process to fractionate bioactive compounds from tomato leaves extract. The paper is logically structurized and clearly presented. However, it resembles more technical report rather than a scientific publication. Fouling phenomena have to be discussed in more detail taking into account membrane hydrophilic-hydrophobic properties and surface roughness parameters.
Comments
- State, please, clearly what is the novelty of the study. Cite and analyze, please, more references on membrane-based processes for extraction of valuable compounds from tomatos or cultivated solanaceous crop species.
ANSWER: Different sentences were added in the manuscript, which are in the following reported to better discuss the obtained results with the ones related to other solanaceous crop and membrane processes, highlithing the novelty of the work:
-) In the introduction
Page 2 line 80: “Various membrane processes have been developed to concentrate and clarify tomato juice [15][16][17][18] or to extract lycopene from tomato peels [19][20]. However, in order to use biomass that does not compete with food, tomato leaves were used for the first time in this work as a starting material for the production of bioactive molecules. Although membrane processes have widely demonstrated their advantages in the pro-cessing of agricultural and food streams, to our knowledge there is little information in the open literature on the type of the membranes, the processes and their performance in the treatment of tomato leaf extracts, and the bioactivity of the processed streams.”
Page 2 line 96: “The challenge is to develop a system capable of recovering bioactive compounds from the original biomass while preserving their structure and function so that their ability to stimulate plant defense is maintained.”
-) In the discussion:
Page 12 line 391 “Similar rejection results against carbohydrates and biophenols were obtained by Conidi et al. [39], using 1 kDa polyamide and polyethersulfone membranes, but starting from an-other solanaceous crop extract (Goji leaves).”
Besides the following new reference were added:
Petrotos, K. B.; Quantick, P.; Petropakis, H. A. study of the direct osmotic concentration of tomato juice in tubular membrane–module configuration. I. The effect of certain basic process parameters on the process performance. J. Membr. Sci. 1998, 150(1), 99-110.
Bottino, A.; Capannelli, G.; Turchini, A.; Della Valle, P.; Trevisan, M. Integrated membrane processes for the concentration of tomato juice. Desalination, 2002 148(1-3), 73-77.
Razi, B.; Aroujalian, A.; Raisi, A.; Fathizadeh, M. Clarification of tomato juice by cross‐flow microfiltration. J. Food Sci. Technol. 2011, 46(1), 138-145.
Bahçeci, K. S.; Akıllıoğlu, H. G.; Gökmen, V. Osmotic and membrane distillation for the concentration of tomato juice: Effects on quality and safety characteristics. Innov. Food Sci. Emerg. Technol. 2015, 31, 131-138.
de Souza, A. L. R.; Gomes, F. D. S.; Tonon, R. V.; da Silva, L. F. M.; Cabral, L. M. C. Coupling membrane processes to obtain a lycopene‐rich extract. J. Food Process. Preserv. 2019, 43(11), 14164.
Yodjun, P.; Soontarapa, K.; Eamchotchawalit, C. Separation of lycopene/solvent mixture by chitosan membranes. J. Met. Mater. Miner. 2011, 21(1).
Conidi, C., Drioli, E., & Cassano, A. (2020). Coupling ultrafiltration-based processes to concentrate phenolic compounds from aqueous goji berry extracts. Molecules, 25(16), 3761.
- Why pH 2.3 is preferable for the extraction?
ANSWER: At the mentioned pH, a highest amount of carbohydrates (0.1 g/gleaves) respect to the other pH values was obtained (page 6 line 219). Besides, the acidic pH permits other advantages e.g avoiding phenols oxidation, milew groth and smell formation. The following sentence was added to better clarify this point together with a new reference:
“Besides, a pH around 2 avoid phenols oxidation and mildew growth” .
The added reference is:
Bazzarelli, F., Poerio, T., Mazzei, R., D’Agostino, N., & Giorno, L. (2015). Study of OMWWs suspended solids destabilization to improve membrane processes performance. Separation and Purification Technology, 149, 183-189.
- Determine, please, and compare membrane water contact angles and use these data to discuss membrane fouling. Is PES membranme hydrophilized or not?
ANSWER:
Yes, PES membrane is a hydrophilized commercial membrane. In order to better clarify this point this information was added in the text in Materials section.
Water contact angles of all the used memebranes were now present in table 1 and the discussion about fouling was implemented modifying the text on page 8 line 301 as follows: “The lower fouling tendency is due to the high hydrophilicity of the RC membrane (water contact angle: 19°±4 [24]) respect to the other membranes (Table 1), which favors low fouling due to protein adsorption, as previously shown by [13][ 32] using other types of biomass as starting materials.”
And on page 11 line 367: “Membranes with the same NMWCO (5 kDa PES and ZrO2) (Figure. 7) showed a comparable tendency to foul, consistent with their hydrophilicity (water contact angle in Table 1). In contrast, analysis of membranes made of the same material (same roughness, 5 kDa: 1.59 nm and [25]1 kDa: 1.30 nm [26]) but with different NMWCO (PES membranes) showed a higher FI for the 1 kDa membrane, in agreement with its lower hydrophilicity (Table 1), resulting in a higher fouling tendency.
Consistent with the flux behavior, the CA membranes showed lower FI (Figure 7) because they have a lower chemical interaction with biomolecules due to a different membrane material and surface chemistry, resulting in a lower fouling tendency.
The following new references were also added
Besides a new paragraph was included in Methods section to describe the water contact angle measurement: “Membrane wettability was measured by static water contact angle measurement (CAM 200 instrument, KSV Instruments, Ltd.). Briefly, water (5 μl) was applied to the membrane with an automatic microsyringe, and each measurement was repeated five times on different membrane pieces.”

Round 2
Reviewer 2 Report
The authors satisfactorily responded to all comments, so the paper can be accepted for publications.